# Environmental and Behavioral Risk Factors for Severe Leptospirosis in Thailand

**DOI:** 10.3390/tropicalmed4020079

**Published:** 2019-05-16

**Authors:** Soawapak Hinjoy, Somkid Kongyu, Pawinee Doung-Ngern, Galayanee Doungchawee, Soledad D. Colombe, Royce Tsukayama, Duangjai Suwancharoen

**Affiliations:** 1Office of International Cooperation, Department of Disease Control, Ministry of Public Health, Nonthaburi 11000, Thailand; 2Bureau of Epidemiology, Department of Disease Control, Ministry of Public Health, Nonthaburi 11000, Thailand; skongyu@gmail.com (S.K.); pawind@gmail.com (P.D.-N.); 3Department of Pathobiology, Faculty of Science, Mahidol University, Bangkok 10400, Thailand; galayanee.dou@mahidol.ac.th; 4Center for Global Health, Weill-Cornell Medical College, 1300 York Ave, New York, NY 10065, USA; soledad.colombe@gmail.com; 5Department of Infectious Disease and Vaccinology, School of Public Health, University of California, Berkeley, CA 94720, USA; roycetsukayama@gmail.com; 6National Institute of Animal Health, Department of Livestock Development, Ministry of Agriculture and Cooperatives, Bangkok 10900, Thailand; dj90705@hotmail.com

**Keywords:** case-control study, leptospirosis, adult, environment, behavior

## Abstract

A nationwide prevention and control campaign for leptospirosis in Thailand has led to a decreased incidence rate, but the mortality and case fatality rates have remained stable. Regarding the limited knowledge of risk factors, a case-control study of the association between environmental and behavioral exposure with severe leptospirosis was implemented to identify the risk factors among adults in Thailand. The study was conducted in 12 hospital-based sites. Hospitalized patients with suspected clinical symptoms of leptospirosis were tested for leptospirosis by culture, loop mediated isothermal amplification (LAMP), real-time PCR, and the microscopic agglutination test (MAT). All participants answered a standardized questionnaire about potential risk factors. Risk factors were identified by univariable and multivariable logistic regression. Of the 44 confirmed cases, 33 (75.0%) presented with severe illness, as determined by clinical criteria, and were categorized as severe cases. Non-severe cases were defined as patients with non-severe symptoms of leptospirosis. Living nearby a rubber tree plantation (adjusted OR 11.65, 95% CI 1.08–125.53) and bathing in natural bodies of water (adjusted OR 10.45, 95% CI 1.17–93.35) were both significantly associated with an increased risk of severe leptospirosis. We recommend designating rubber plantations in Thailand as high-risk zones and closely monitoring hospitalized patients in those areas.

## 1. Introduction

Leptospirosis is a zoonotic disease with a worldwide distribution caused by spirochetes of the genus *Leptospira*. Humans are rarely chronic carriers and are considered accidental hosts. Two forms of disease are found in humans: icteric and anicteric. Anicteric leptospirosis is a mild disease and typically self-limited, while the icteric form is more severe. Icteric leptospirosis, which often progresses rapidly, occurs in 5–10% of all infected patients, and is associated with severe jaundice, liver failure, renal failure, pulmonary involvement, and even death [1,2].

In 2000, epidemics of leptospirosis occurred following severe flooding in the northeastern and southern regions of Thailand. There were more than 14,000 reported cases, with an incidence rate of 23.1 per 100,000 population and a case fatality rate of 2.5% [3]. An intensive prevention and control campaign including integrated pest control was implemented nationwide in response to the outbreaks. Campaigns for intensive and systematic rodent control, education of proper sanitation and hygiene practices, and promoting theuse of boots and gloves during farming work have been implemented [4] and repeatedly promoted by the Ministry of Public Health (MoPH) of Thailand led to an observed decrease in the incidence rate to 5.9 per 100,000 population [3]. A study by Thipmontree et al. [5] observed that although thenumber of leptospirosis cases in Northeast Thailand had decreased since 2001, the case fatality rate of severe leptospirosis had increased from 2001 to 2012 with severe lung hemorrhage as a major cause of death among leptospirosis cases. Another study by Niwattayakul et al. [6] from 1999 to 2000 showed the complicated clinical manifestations of severe leptospirosis in Thailand, revealing thatthe major causes of death included pulmonary complications, renal failure, and sepsis.

Although epidemiological, serological, and molecular research have increased our understanding of leptospirosis, the association between severe leptospirosis and potential environmental and behavioral risk factors has not been adequately studied in Thailand.

The major factors influencing complications of leptospirosis include the serovar of *Leptospira* spp. involved, the susceptibility of the host, the host environment, and the behavior of the host [7]. In addition, the severity of clinical signs has been shown to be positively associated with the number of skin abrasions on the host [8]. Ganoza et al. [9] identified *Leptospira* spp. in environmental samples and showed that severe leptospirosis in the Peruvian Amazon was associated with higher concentrations of pathogenic *Leptospira* spp. at the sites of exposure and transmission. Environmental factors and host behavior could therefore be significant factors in the severity of leptospirosis in Thailand. Mwachui et al. conducted a systematic review of studies on the environmental risk factors of human leptospirosis, which included 14 studies from Southeast Asia [10]. One study in Lao PDR did not find an association of an increased risk of infection with contact with environmental streams, contradicting the increased risk of infection found in all of the other studies [11].

Knowledge of risk factors for severe leptospirosis can improve prevention and control measures to mitigate the incidence of severe cases. Thus, we aimed to identify the environmental and behavioral risk factors associated between severe and non-severe leptospirosis among adults in Thailand. 

## 2. Materials and Methods

### 2.1. Study Design

We conducted a case-control study in 12 hospitals across Thailand. These hospitals were located in areas of high incidence of human leptospirosis cases in Thailand according to surveillance data from the national disease surveillance system [3].The study area included four hospital sites located in northern Thailand including Nan, Tha Wang Pha, Chaloem Phra Kiat, and Chiang Klang Hospitals; three sites in northeastern Thailand including Mahasarakham, Kamalasai, and Uthumphonphisai Hospitals; and five sites in southern Thailand including Ranong, Suratthani, Nakhon Si Thammarat, Phatthalung, and Trang Hospitals, as shown in Figure 1.

There were seven provincial hospital sites and five district hospitals. We enrolled all patients aged 18–60 years old presenting at the hospitals between May and September 2017 with suspected leptospirosis, sepsis, acute renal failure, pneumonia, fever of an unidentified source, or suspected dengue. Patients residing in the study areas for less than two weeks were excluded from the study. All suspected patients provided informed consent to provide blood samples for leptospirosis screening by polymerasechain reaction (PCR) assay, culture, and loop-mediated isothermal amplification (LAMP) specific for the 16S ribosomal RNA gene (rrs) of pathogenic and intermediate clusters of *Leptospira* species.

The *Leptospira* serogroup was determined by a microscopic agglutination test (MAT) and all confirmed *Leptospira* samples were defined assevere cases or non-severe cases. The severe subjects consisted of patients who were positive for *Leptospira* spp.inat least one test and admitted at the hospital with at least one of the following signs: conjunctival suffusion, jaundice with alanine aminotransferase (ALT) and aspartate aminotransferase (AST) three times above the normal level (>120 U/L), jaundice with serum creatinine >133 μmol/L or blood urea nitrogen >26.8 mmol/L, pulmonary hemorrhage, respiratory insufficiency with a respiratory rate >30 breaths/min, use of supplemental oxygen therapy, and/or death. The non-severe subjects consisted of patients who were admitted at the hospital without any of the aforementioned clinical signs and were positive for *Leptospira* spp. in at least one test. Data on demographic factors and risk factors included a history of clinical signs, dietary habits, exposure to water, work history, exposure to animals and rodents, presence of animal farms within 1 km surrounding the living areas, sharing a natural water resource with animals, personal protective equipment practices, and geographic areas such as their households and working places’ proximity to communities, rural areas, rice farms, rubber plantations and oil palm trees, and sugarcane, cassava, and fruit gardenswere recorded using a standardized questionnaire. All interviews and the collection of blood specimens were conducted in the hospitals. 

### 2.2. Laboratory Analysis

All participants provided a blood sample that was placed into a tube of semisolid culture medium, and immediately transported to a laboratory. Second blood samples were transferred into two tubes. One sample was treated with EDTA to extract for PCR and LAMP assays, and the other sample was placed into a tube without EDTA, in order to separate the serum for MAT. All samples were tested at the National Institute of Animal Health (NIAH) in Bangkok, Thailand. Confirmed leptospirosis was defined as a positive result in any of the following tests: polymerasechain reaction (PCR)assay, culture, or loop-mediated isothermal amplification (LAMP) specific for the 16S ribosomal RNA gene (rrs) of pathogenic and intermediate clusters of *Leptospira* species.

### 2.3. Real-Time PCR

Real-time PCR was carried out by targeting the 241 bp region of the *lipL32* gene, which encodes the major outer membrane of pathogenic *Leptospira* spp. [12,13,14]. The master mix (20 µL) was prepared from Fast Start DNA probe master (Roche, Switzerland), primer and probes (LipL32-45F, LipL32-286R, Probe-189), and 5 µL of DNA primer. Amplification was done by the LightCycler^®^ nano instrument (Roche Diagnostics International Ltd., Rotkreuz, Switzerland) at 45 cycles. Positive samples of the *lipL32* gene were molecularly and genetically identified.

### 2.4. Leptospira Loop–Mediated Isothermal Amplification Method (LAMP)

DNA in all blood samples was extracted using the QIAamp DNA Mini Kit (Qiagen, Netherlands). DNA samples were kept at −20 °C until use. The LAMP assay was carried out using primers and conditions previously described by Suwancharoen et al. [15]. This method used 10 ng of extracted DNA and a 25 µL reaction mixture containing 1.0 µM outer primers F3 and B3, 1.6 µM inner primers FIP and BIP, 0.8 µM loop primer LB, 1X Thermopol^®^ reaction buffer (New England Biolabs, MA, USA), 4 mM MgSo_4_ (New England Biolabs, MA, USA), 1 M Betaine (Sigma, MO, USA), 0.4mMdNTPs (New England Biolabs, MA, USA), 0.5 mM MnCl, 25 µM Calcein, and 8 units *Bst* DNA polymerase (New England Biolabs, MA, USA). The reaction mixture was incubated at 61 °C for 90–120 min in a heating block and then heated at 80 °C for 2 min to terminate the reaction. The presence of *Leptospira* DNA was defined on the basis of detecting a green color with the naked eye. 

### 2.5. Leptospira spp. Culture

The blood samples were cultured in EMJH semi–solid media (Johnson and Harris modification of the Ellinghausen and McCullough medium, Difco, Baltimore, USA) at 30 °C for 12–16 weeks. The presence of leptospires was examined by dark field microscopy to observe the motility and the characteristic thin helical structures with prominent hooked ends [16]. The suspected samples were further sub-cultured in the EMJH liquid media and purified with a 0.2 µm–pore–size membrane filter to remove the contaminants.

### 2.6. Serum Detection

The MAT was performed with a panel of 24 reference serovars based on the standard method described by the WHO/FAO/OIE Collaborating Centre for Reference and Research on Leptospirosis [17,18]. The panel of antigens was representative of 23 pathogenic serogroups and one non-pathogenic serogroup. The serogroups used in the study were the *L. interrogans* serogroups Australisserovar Bratislava, Autumnalis, Ballum, Bataviae, Canicola, Celledoni, Cynopteri, Djasiman, Grippotyphosa, Hebdomadis, Icterohaemorrhagiae, Javanica, Louisiana, Manhao, Mini, Panama, Pomona, Pyrogenes, Ranarum, Sarmin, Sejroe, Shermani, Tarassovi, and *L. biflexa* serovar Patoc. The panel of antigens was obtained from the WHO/FAO/OIE Collaborating Centre for Reference and Research on Leptospirosis, Western Pacific Region, Brisbane, Queensland, Australia. A MAT-positive result was set at a titer of ≥1:400.

### 2.7. Statistical Methods

All data were entered into Microsoft Access version 2007. Data management and all analyses were performed using Epi-Info, version 7. Differences in demographic and clinical variables between the severe cases and non-severe cases were described using a chi-squared test and by Fisher’s Exact test. Univariable and multivariable logistic regressions were performed to estimate the odds ratios (OR) and 95% confidence intervals (CI) of various potential risk factors for severe leptospirosis. In order to investigate whether any association was caused by confounding factors, multiple logistic regression analysis was performed. The model was reduced in a backward elimination procedure. A 10% change in the coefficient was considered evidence of possible confounding. The sex of the patient and the presence of underlying disease(s) were considered the most important potential confounders, and were forced into the final model, regardless of their univariable significance. Adjusted OR’s and 95% CI’s were also calculated. A 2-sided p-value of less than 0.05 was considered statistically significant in all analyses.

### 2.8. Ethics

All eligible participants were invited to participate in the study through a letter informing them of the requirements and each participant signed a written informed consent document before enrolment. All study procedures were reviewed and approved by the Ethics Committee for Research in Human Subjects, Department of Disease Control, Ministry of Public Health, Thailand (FWA 00013622) on 26 December 2016. 

## 3. Results

Of the 173 suspected patients, 44 (25.4%) had a laboratory-confirmed *Leptospira* infection. There were no signs of growth by culture. Of the 44 patients with a confirmed *Leptospira* infection, 28(63.6%) were cases with typical clinical leptospirosis, 10 (22.8%) were cases with fever of an unidentified source, 2 (4.5%) were cases with acute renal failure, 2 (4.5%) were cases with pneumonia, 1 (2.3%) was a case with sepsis, and 1 (2.3%) was a case of suspected dengue. The case fatality rate was 2.3% (one death of 44 confirmed patients). Of the 44 laboratory-confirmed cases, 33 (75.0%) presented with severe illness and were categorized as the case group while 11 (25.0%) presented with non-severe symptoms and were categorized as the control group. There were significant differences in terms of clinical factors includingchill, conjunctival suffusion, jaundice, breathlessness, and oliguria between the two groups as shown in Table 1.

The three most common serogroups identified in infected subjects were Australis (22/38–57.8%), Icterohaemorrhagiae (3/38–7.9%), and Pyrogenes (3/38–7.9%) as shown in Table 2. None of the non-severe leptospirosis had positive titers against Icterohaemorrhagiae, which were identified based on the highest MAT titers.

When exploring the behavioral and environmental risk factors for leptospirosis (Table 3), living nearby rubber tree plantations as well as bathing in natural bodies of water within two weeks before the illness were significantly associated with an increased risk of severe leptospirosis compared to non-severe leptospirosis after univariable analyses (OR 12.00, 95% CI 1.37–104.77 and OR 7.25, 95% CI 1.43–36.69, respectively). Severe cases reported bathing in stagnant water (41.9%), slowly-flowing water (29.0%), and mud (29.0%).

After the multivariable analyses (Table 4) and adjusting for sex and underlying diseases, living nearby rubber tree plantations still had the strongest association with severe leptospirosis in the multivariate model (Adjusted OR 11.65, 95% CI 1.08–125.53). Bathing in natural bodies of water two weeks before illness (Adjusted OR 10.45, 95% CI 1.17–93.35) was an additional independent risk factor for severe leptospirosis.

## 4. Discussion

Living near a rubber plantation and bathing in natural bodies of water were the main risk factors for developing severe leptospirosis in our study population. This study is the first, to our knowledge, to investigate environmental and behavioral risk factors associated with severe leptospirosis in Thailand. 

Living nearby rubber tree plantations and bathing in natural bodies of water two weeks before the onset of illness are activities associated with agricultural and outdoor occupations, which are typically associated with the transmission and infection of leptospirosis [19]. These two behaviors have previously been shown to be associated with human leptospirosis infections [20,21]. The reason for why they are associated with severe leptospirosis in our study was probably due to the high concentration of the specific *Leptospira* serovars found in rubber tree plantations and natural bodies of water.

In our study, severe cases were more likely than non-severe cases to be infected with Icterohaemorrhagiae serogroups. Infectious *Leptospira* serogroups have been shown to be associated with the severity of disease in humans [22,23]. In both Guadeloupe and Martinique, the *Leptospira interrogans* serogroup Icterohemorrhagiae was linked to severe outcomes [2,24]. From a study by Chadsuti et al. [25], the most predominant serovars in human serum samples under suspicion of either severe or non-severe cases leptospirosis were Shermani, followed by Bratislava, Panama, and Sejroe.

In addition, rodents are natural carriers of both serogroups Sejroe and Icterohaemorrhagiae in Thailand [26,27,28,29,30]. Rubber tree plantations are areas with a high likelihood of rat infestation [31]. Palm oil and rubber plantations are very similar in many aspects such as cultivation practices [32]. Many rat species are found in palm oil plantations in Peninsular Malaysia [33,34]. A study on palm oil plantations showed that the rat population varied among plots, ranging from 84 to 564 rats ha^−1^ (mean of 298 rats ha^−1^) [35]. In addition to a high density of rat populations in palm oil and rubber plantations, a favorable tropical climate and surface environment conditions in those plantations means a long survival time for the pathogen [36,37]. These *Leptospira* pathogens from the urine of infected rats may contaminate soil and water in rubber plantations, and may also spread to nearby areas. 

In natural bodies of water, it is likely that the concentration of highly pathogenic *Leptospira* ishigh, thus the inoculation dose for patients was high, leading to severe leptospirosis. Rodents are likely to contaminate ponds or stream water [38,39] where there is evidence of higher concentrations of *Leptospira* organisms*. Leptospira* survives better in stagnant river water than in rainwater or underground water [5] due to its ability to survive in water and soil [40]. Among our study population, severe subjects reported behavior of bathing in stagnant water (41.9%), slowly-flowing water (29.0%), and mud (29.0%), which reinforces thehypothesis that patients were exposed to a high concentration of *Leptospira* organisms through contact with stagnant water.

Our results are to be interpreted in light of some limitations. The number of confirmed *Leptospira* infections was relatively small (*n* = 44 cases). In addition, our study is likely to be affected by selection bias. The probability of hospitalization of severe cases and non-severe cases might differ and could be influenced by the exposure. Non-severe cases of leptospirosis were the most common form of the disease, accounting for approximately 90% of the cases. Only 10% of non-severe cases were admitted to the hospital [41]; therefore, the non-severe subjects in this study may not be an accurate representation of the non-severe casesin the target population. Additionally, environmental exposure and behavior could differ between severe cases and non-severe cases and influence the rates of hospitalization of our study population. Information bias might have been present due to the definition of severe leptospirosis, which was confirmed in cases by laboratory methods and patients admitted at the hospital with at least one of the following signs: conjunctival suffusion, and others. The marker of conjunctival suffusion as a symptom of severe leptospirosis was from pragmatic and referred experiences of medical management in the study areas, while other studiesdid not include conjunctival suffusionin the definition of severe leptospirosis [2,42]. Another risk to the severity of leptospirosis, time of onset to initiation of antibiotherapy, was not measured among the patients in this study due to invalid responses. The findings of living nearby rubber tree plantations and bathing in natural bodies of water as high risk factors are important to guide policy-making agencies for justifying campaigns to control leptospirosis transmission. Specific recommendations regarding the designation of rubber tree plantations in Thailand as high-risk zones should include the close monitoring of any suspected leptospirosis cases among patients with fever of an unidentified source, strengthening disease surveillance, and raising awareness among physicians to implement early treatment for preventing complications. General health promotion practices such as wearing rubber shoes or gloves during bathing or soaking in water should also continue. Rodent control measures and improvement of environmental exposures at home, rubber tree plantations, and in the workplace are necessary to prevent both severe and non-severe cases leptospirosis in endemic regions. Finally, further studies are needed to determine the prevalent serogroups, their survival in the environment, and their association with the severity of disease in Thailand. More research is also needed to determine whether the exposure is occurring at the rubber plantations or if the proximity to rubber plantations is indicative of other potential exposures such as rural and isolated lifestyles associated with exposure.

## Figures and Tables

**Figure 1 tropicalmed-04-00079-f001:**
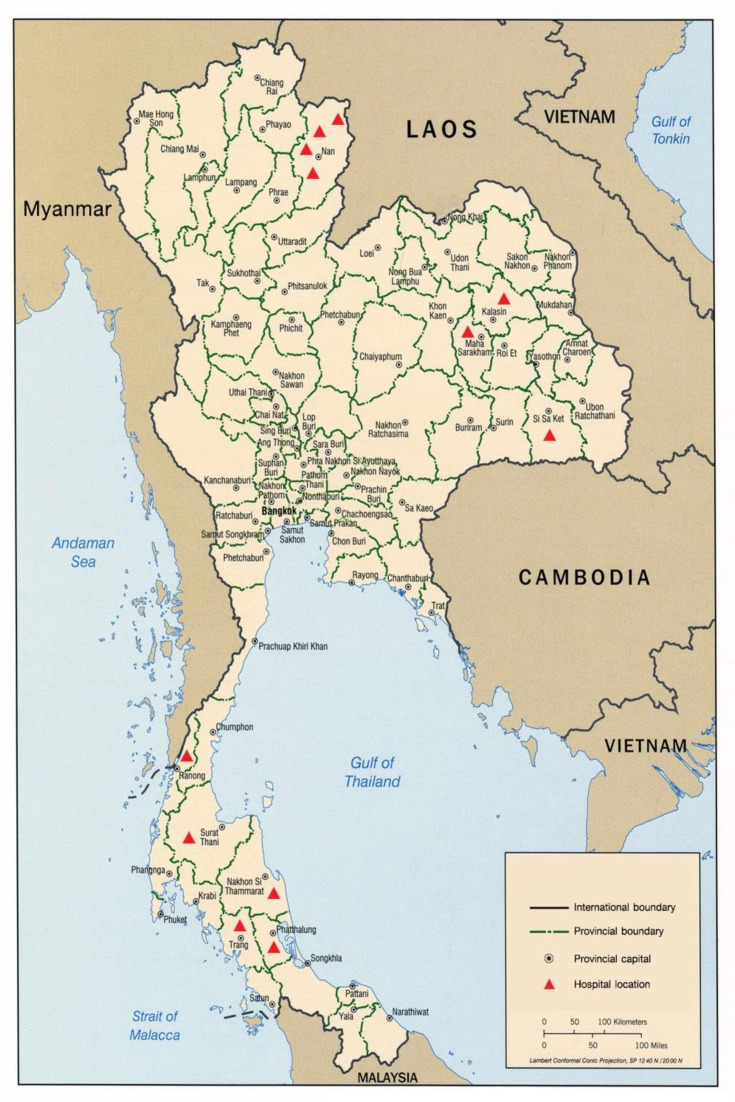
Location of the 12 hospital-based sites in Thailand, 2017.

**Table 1 tropicalmed-04-00079-t001:** Demographic and clinical characteristics of confirmed leptospirosis from 12 hospital-based sites in Thailand, 2017.

Characteristics	Severe Leptospirosis n (%) or Median (IQR) *N* = 33	Non-Severe Leptospirosis n (%) or Median (IQR) *N* = 11	*p*-Value
**Demographics and behavior**			
Age	44	47	0.26
Male gender	27 (81.8)	7 (63.6)	0.21
Alcohol consumption	25 (75.8)	7 (63.6)	0.43
Cigarette smoking	21 (63.6)	5 (45.5)	0.29
**Clinical presentation**			
Acute fever	31 (96.9)	10 (90.9)	0.42
Chill	29 (87.9)	5 (45.5)	0.004
Myalgia	26 (78.8)	7 (63.6)	0.31
Calf pain	23 (69.7)	4 (36.4)	0.08 *
Severe headache	25 (75.8)	6 (54.6)	0.18
Stiff neck	11 (34.5)	2 (18.2)	0.46
Conjunctival suffusion	16 (48.5)	0	0.003 *
Jaundice	19 (57.6)	0	0.0009 *
Dry cough	15 (45.5)	4 (36.4)	0.73 *
Cough blood-tinged sputum	5 (15.2)	0	0.31 *
Hemoptysis	1 (3.0)	0	1.00 *
Breathlessness	18 (54.6)	1 (9.1)	0.01 *
Vomiting blood	2 (6.1)	0	Not valid
Melena	2 (6.1)	1 (10.0)	0.56 *
Dark urine	18 (54.6)	1 (9.1)	Not valid
Oliguria	13 (39.47)	0	0.02 *
Having underlying diseases such as diabetes mellitus, hypertension, etc.	11 (33.4)	4 (36.4)	1.00 *
Days between date of onset and date of admission	4.0 (4.0)	4.0 (3.0)	0.62
Death	1 (3.0)	0	1.00 *

* Statistics by Fisher Exact test.

**Table 2 tropicalmed-04-00079-t002:** Serogroups (with the highest titer by a microscopic agglutination test (MAT)) of confirmed leptospirosis from 12 hospital-based sites in Thailand, 2017.

Infecting Serogroups	Severe Leptospirosis n (%) *N* = 29 *	Non-Severe Leptospirosis n (%) *N* = 9 **	Total n (%) *N* = 38
Australis	14 (48.4)	8 (88.9)	22 (57.8)
Icterohaemorrhagiae	3 (10.3)	0	3 (7.9)
Pyrogenes	2 (6.9)	1 (11.1)	3 (7.9)
Sejroe	2 (6.9)	0	2 (5.3)
Autumnalis	2 (6.9)	0	2 (5.3)
Djasiman	2 (6.9)	0	2 (5.3)
Hebdomadis	2 (6.9)	0	2 (5.3)
Bataviae	1 (3.4)	0	1 (2.6)
Grippotyphosa	1 (3.4)	0	1 (2.6)

* Four cases had no information on the serogroup due to testing positive formore than one pathogenic serogroup with the same high titer; ** Two cases had negative results by the MAT.

**Table 3 tropicalmed-04-00079-t003:** Odds ratios and 95% confidence intervals of risk factors for severe leptospirosis from 12 hospital-based sites, Thailand 2017.

Variable	Severe Leptospirosis n (%) *N* = 33	Non-Severe Leptospirosis n (%) *N* = 11	OR (95% CI)
**Reservoirs**			
Contact with animals > 4 h/day	3 (13.6)	1 (11.2)	1.26 (0.11–14.05)
Having rat cage(s) in house	7 (21.2)	4 (36.4)	0.47 (0.11–2.08)
Rat feces sighted at food storage	14 (42.4)	6 (54.6)	0.61 (0.16–2.42)
Rats sighted at field sites	26 (78.8)	9 (81.8)	0.83 (0.14–4.73)
**Residential sanitary condition**			
Household flooded in the rainy season	11 (33.4)	4 (36.4)	0.88 (0.21–3.64)
Not having food and water storage containers	6 (18.2)	3 (30.0)	0.52 (0.10–2.61)
Never using boots	9 (28.1)	1 (10.0)	3.52 (0.39–31.95)
Living nearby rubber tree plantations	18 (54.6)	1 (9.1)	12.00 (1.37–104.77)
Presence of pond/canal < 1 km from house	28 (84.9)	9 (81.8)	1.25 (0.21–7.56)
Presence of cattle < 1km from house	23 (71.9)	7 (63.6)	1.46 (0.34–6.22)
Sharing water resources with livestock	24 (77.4)	7 (87.5)	0.49 (0.05–4.69)
Wastedisposal sites nearby house	10 (30.3)	1 (9.1)	4.35 (0.49–38.68)
**Exposure to contaminated sources**			
Frequent exposure to bodies of water	24 (72.7)	7 (63.6)	1.52 (0.36–6.48)
Bathing in water ≤ 2 weeks before illness	29 (87.9)	5 (50.0)	7.25 (1.43–36.69)
Having a wound ≤ 2 weeks before illness	16 (50.0)	5 (45.5)	1.20 (0.30–4.74)
**Occupation**			
Worked in the wet-rice fields ≤ 2 weeks before illness	25 (75.8)	7 (63.6)	1.79 (0.41–7.72)
Fished ≤ 2 weeks before illness	16 (48.5)	2 (18.2)	4.24 (0.79–22.7)
Worked in sewage ≤ 2 weeks before illness	1 (3.0)	1 (9.1)	0.31 (0.02–5.46)
Hunted rats at night ≤ 2 weeks before illness	1 (3.0)	1 (9.1)	0.31 (0.02–5.46)

**Table 4 tropicalmed-04-00079-t004:** Unadjusted and adjusted odds ratios and 95% confidence intervals risk factors for leptospirosis.

Risk Factors	Severe Leptospirosis n (%) *N* = 33	Non-severe Leptospirosis n (%) *N* = 11	Unadjusted OR (95% CI)	Adjusted OR (95% CI)
Bathing in natural bodies of water 2 weeks before illness	29 (87.9)	5 (50.0)	7.25 (1.43–36.69)	10.45 (1.17–93.35)
Living nearby rubber tree plantations	18 (54.6)	1 (9.1)	12.00 (1.37–104.77)	11.65 (1.08–125.53)
Male	27 (81.8)	7 (63.6)	2–57 (0.57–11.69)	3.61 (0.40–32.23)
Having underlying disease(s)	11 (33.4)	4 (36.4)	0.88 (0.21–3.64)	3.81 (0.39–37.75)

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
