# Peer review of "Environmental and Behavioral Risk Factors for Severe Leptospirosis in Thailand"

_tropicalmed, 2019, doi:10.3390/tropicalmed4020079_

Round 1
Reviewer 1 Report
This article aims to identify the risk factors of severe letpospirosis patients through a case-control study in 12 hospitals, conducted on 173 patients (including 33 severe cases and 11 with non-severe symptoms and categorized as controls).
Introduction
The introduction raises the issue in Thailand but is brief on the literature review of other studies that have considered environmental and behavioural factors.
Especially the systematic review of Mwachui et al. In 2015 is expected, since this article deeply looked at 64 articles. Some of the cited articles may also be of a major interest for this introduction.
Mwachui, M. A., L. Crump, R. Hartskeerl, J. Zinsstag, et J. Hattendorf. 2015. « Environmental and Behavioural Determinants of Leptospirosis Transmission: A Systematic Review ». Plos Neglected Tropical Diseases 9 (9). https://doi.org/10.1371/journal.pntd.0003843.
Line 55: need citation
Material and Methods:
The authors should name the 12 hospitals (they are named in Acknowledgements but this could be first in M&M) and present their location. A map with the location of each of the 12 hospitals would be the best to understand how they are distributed throughout Thailand.
The authors should also present more precisely the questionnaire. If possible, this questionnaire could be provided as supplementary material. For instance, we learn from the results that two questions were about “living nearby rubber trees plantations” and the presence of “pond/canal < 1km from house” but we do not know if other landscapes features were investigated (such as proximity to rice fields, proximity to forests, house isolated or within a village, etc.)
I don't think this study is strictly speaking a case-control study. In my opinion, controls should be people who have the same behavioural and living conditions ("environmental") characteristics as patients. This would require a very large number of them, taking into account the incidence of leptospirosis. Here, I find that it is more a comparison of two groups: one with severe symptoms and the other without.
For statistical analyses, the authors use a chi-squared test to assess differences between cases and controls. This test should be applied under the Cochran’s rule, i.e. when there are at least 5 individuals in each category. Since the number of cases and controls are low, there are often situations with less than 5 individual and in that case, they may use the Fisher test.
Results:
Table 1:
- precise the p-value of what (here, it might be the chi-squared test)
- how this p-value is calculated for the age? 44 and 47 might be the average age of patients. In that case, no decimal should be given, since data might have been recorded as integer. Also what is the meaning of the percentage given in brackets for age?
- Why the clinical presentation are not tested (no p-value given). It might be possible for a few of them.
- Why is the meaning of “--“. If controls were not tested, are we sure that all the 33 cases were also tested? If it means 0, it should be written 0
Table 3:
For instance, when assessing the difference between cases and controls regarding the question “Living nearby rubber tree plantations”, there are 18 vs 15 for cases and 1 vs 10 for controls. Since 1 < 5, they should use a Fisher test which would also indicate that the difference is significative (p-value < 0.05)
Discussion
Lines 221 to 226: the authors should first compare the serovar test results with those previously performed in Thailand to see if these results are new or in agreement.
Line 228: Do the authors have a reference for this assertion “Rubber tree plantations are areas with a high likelihood of rat infestation». Rubber tree plantations have also a low biodiversity and may have less food for rodents than natural environments (https://onlinelibrary.wiley.com/doi/full/10.1111/conl.12170).
Line 236: correct the repetition of “that”
Line 246: Give more information on Berkson’s bias, which is a bias in the admission rate
Tests indicate a link between proximity to rubber plantations and leptospirosis. However, this does not mean that populations are most exposed in rubber plantations, and the authors should specify and temper this result. This proximity may simply be an indicator of exposure to leptospirosis. This proximity can also refer to more rural populations, living more isolated, who may be more in contact with water and soil, or more in contact with wildlife, or who have unhygienic living conditions, etc.
Instead of environmental factors, I would rather talk about factors related to the place of residence and living conditions, because this study is not an environmental study in the strict sense of the term. An environmental study would involve a more accurate description of the environment around patients.
This article is interesting because it is indeed important to understand the exposure factors. It therefore requires a major revision and I propose here some ways to address it.
Author Response
Introduction
The introduction raises the issue in Thailand but is brief on the literature review of other studies that have considered environmental and behavioural factors.
Response. Already added in the context (between line 78-82) and showed in a reference #10 and 11
Especially the systematic review of Mwachui et al. In 2015 is expected, since this article deeply looked at 64 articles. Some of the cited articles may also be of a major interest for this introduction.
Response One article from a total 64 articles cited by Mwachui et al. have been already cited in this manuscript as shown in an author name of Ganoza et al.
Mwachui, M. A., L. Crump, R. Hartskeerl, J. Zinsstag, et J. Hattendorf. 2015. « Environmental and Behavioural Determinants of Leptospirosis TrResponsemission: A Systematic Review ». Plos Neglected Tropical Diseases 9 (9). https://doi.org/10.1371/journal.pntd.0003843.
Line 55: need citation
Response Already cited in the context as a reference #7
Material and Methods:
The authors should name the 12 hospitals (they are named in Acknowledgements but this could be first in M&M) and present their location. A map with the location of each of the 12 hospitals would be the best to understand how they are distributed throughout Thailand.
Response Already added in line between 94 and 97 and displayed as shown in figure 1
The authors should also present more precisely the questionnaire. If possible, this questionnaire could be provided as supplementary material. For instance, we learn from the results that two questions were about “living nearby rubber trees plantations” and the presence of “pond/canal < 1km from house” but we do not know if other landscapes features were investigated (such as proximity to rice fields, proximity to forests, house isolated or within a village, etc.)
Response Already added in line between 122 and 126
I don't think this study is strictly speaking a case-control study. In my opinion, controls should be people who have the same behavioural and living conditions ("environmental") characteristics as patients. This would require a very large number of them, taking into account the incidence of leptospirosis. Here, I find that it is more a comparison of two groups: one with severe symptoms and the other without.
Response The case subjects consisted of patients who were positive for Leptospira spp. and having a sign of severe leptospirosis. The control subjects consisted of patients who were positive for Leptospira spp. and without having a sign of severe leptospirosis. Both the case and control subjects were from the hospital-based population. There were seven provincial hospital sites and five district hospitals. Most of the people in Thailand prefer to go the hospitals nearby their households more than admitting at a hospital across the province. None of the sites were located in megacities of Thailand. Therefore, the cases and controls from our hospital sites mostly shared similar rural environments..
For statistical analyses, the authors use a chi-squared test to assess differences between cases and controls. This test should be applied under the Cochran’s rule, i.e. when there are at least 5 individuals in each category. Since the number of cases and controls are low, there are often situations with less than 5 individual and in that case, they may use the Fisher test.
Response Already corrected in line 185 and results
Results:
Table 1:
- precise the p-value of what (here, it might be the chi-squared test)
Response It's the chi-squared test and already revised to use a Fisher Exact test if At least one cell has expected size <5
- how this p-value is calculated for the age? 44 and 47 might be the average age of patients. In that case, no decimal should be given, since data might have been recorded as an integer. Also, what is the meaning of the percentage given in brackets for age?
Response Age of 44 and 47 are medians of each group. The decimal and percentage in brackets have been deleted
- Why the clinical presentation is not tested (no p-value is given). It might be possible for a few of them.
Response Already tested and stated in the table 1
- Why is the meaning of “--“. If controls were not tested, are we sure that all the 33 cases were also tested? If it means 0, it should be written 0
Response Replaced with 0
Table 3:
For instance, when assessing the difference between cases and controls regarding the question “Living nearby rubber tree plantations”, there are 18 vs 15 for cases and 1 vs 10 for controls. Since 1 < 5, they should use a Fisher test which would also indicate that the difference is significative (p-value < 0.05)
Response We gave a value of 95% confidence interval for all risk factors to indicate the difference between the two groups
Discussion
Lines 221 to 226: the authors should first compare the serovar test results with those previously performed in Thailand to see if these results are new or in agreement.
Response Already added in line between 276 and 279
Line 228: Do the authors have a reference for this assertion “Rubber tree plantations are areas with a high likelihood of rat infestation». Rubber tree plantations have also low biodiversity and may have less food for rodents than natural environments (https://onlinelibrary.wiley.com/doi/full/10.1111/conl.12170).
Response Already added a reference in line 284 - the author addressed that "rats are invasive pests in oil palm plantations"
Line 236: correct the repetition of “that”
Response Already corrected
Line 246: Give more information on Berkson’s bias, which is a bias in the admission rate
Response Use "selection bias" instead of Berkson's bias to explain the sentence of "Only 10% of non-severe cases were admitted to the hospital; therefore, the non-severe subjects in this study may not be an accurate representation of the non-severe cases in the target population."
Tests indicate a link between proximity to rubber plantations and leptospirosis. However, this does not mean that populations are most exposed in rubber plantations, and the authors should specify and temper this result. This proximity may simply be an indicator of exposure to leptospirosis. This proximity can also refer to more rural populations, living more isolated, who may be more in contact with water and soil, or more in contact with wildlife, or who have unhygienic living conditions, etc.
Instead of environmental factors, I would rather talk about factors related to the place of residence and living conditions, because this study is not an environmental study in the strict sense of the term. An environmental study would involve a more accurate description of the environment around patients.
Response Further explanation was described at the end of the discussion.
Reviewer 2 Report
The authors have tried to highlight environmental and behaviour risk factors which is relevant and timely. Underlying disease condition or co-morbid conditions were taken into account which is expected in endemic countries. Occupational health hazard has been analysed. Limitations of the study has been declared. I have following comments and suggestions for your consideration;
The basis of sampling of hospitals should be clarified.
It will be good to explain changing epidemiological pattern of clinical manifestation of leptospirosis in Thailand and other countries which lead to severe, complicated cases such as hepato-renal failure, multi-organ failure and ARDS.
More literature review is needed to explain good practices and lesson learnt on public health interventions for prevention and control of leptospirosis in Thailand. There are many clinical studies done on co-morbidity and complicated cases of leptospirosis in Thailand which should be discussed.
Author Response
The basis of sampling of hospitals should be clarified.
Response Already added in line between 91 and 93
It will be good to explain changing the epidemiological pattern of clinical manifestation of leptospirosis in Thailand and other countries which lead to severe, complicated cases such as hepato-renal failure, multi-organ failure, and ARDS.
Response Already added in line between 54 and 60
More literature review is needed to explain good practices and lesson learned on public health interventions for the prevention and control of leptospirosis in Thailand. There are many clinical studies done on co-morbidity and complicated cases of leptospirosis in Thailand which should be discussed.
Response Already added in line between 50 and 54
Reviewer 3 Report
The MS "Environmental and behavioral risk factors for severe leptospirosis in Thailand" by Soawapak Hinjoy and colleagues describes a study comparing severe cases of leptospirosis to non-severe cases. The design is sound (yet could be made clearer), the methods are appropriate (although interpretation must be revised) and the results are convincing and most importantly prone to guide Public Health.
In the MS as it currently stands, there are many errors and imprecise statements that need to be addressed.
My major comments are as follow:
1. If my understanding is correct, the design is a comparison of severe leptospirosis cases to non-severe leptospirosis cases. If so, the M&M section must be corrected, since controls must also have a positive test for leptospirosis by any test. In lines 86-87, the statement is that controls were "negative by all tests".
2. The interpretation of MAT results is largely incorrect and must be completely revised and corrected. First, MAT is a not a serovar-specific technique, but has only some degree of serogroup-specificity. Therefore, results should be presented as serogroups. The text and table headings and legends need only to change serovar to serogroup and Bratislava (serovar) to Australis (corresponding serogroup), because all other names are eponymous serogroups. In the interpretation of MAT results, (Table 2), the total number cannot be higher than the number of patients (=44), because MAT can either
a. point to one putative infecting serogroup (the one giving the highest titre)
b. provide no information on the serogroup, if more than one pathogenic serogroup have the same high titre.
Consequently, the sentence (lines 184-185) stating that one patient was infected with a mixture of seven serovars is meaningless and must be removed.
3. The criteria used to define severity should be discussed and compared with the criteria used in similar studies. In another study (PMID 23326614), severe cases were those who died or required vasoactive amines, respiratory support or dialysis. Here, conjonctival suffusion alone or acute hepatic damage (ALAT/ASAT) are considered as markers of severity. Also the findings should be compared to former studies. Here too, PMID 23326614 found that cigarette smoking was a risk (not found here) as was serogroup Icterohaemorrhagiae (not considered here, but could be considered once MAT data is corrected). A major contributor to severity through several studies is the time from onset to antibiotherapy, which is not considered but should be discussed.
Minor comments are listed below:
· The introduction could also mention Severe Pulmonary Haemorrhage Syndrome, a very severe form of leptospirosis. (in the 1st paragraph by line 45?).
· Line 50, the sentence "In return, the incidence rate decreased" is surprising, since the causality was not clearly demonstrated, at least to my knowledge. “In return” should be deleted.
· If the design is comparing severe cases to non-severe cases, the phrasing could be improved. e.g. lines 62-63 "... risk factors associated with leptospirosis severity". Also in text and once the Case-control design is presented, the phrasing could use "severe cases" and "non-severe cases" instead of "cases" and "controls".
· The lab methods are not always clear.
o Line 94-95, the blood is collected in a tube of semi-solid culture medium before separation of the serum? Why a culture medium and if so, what culture medium?
o Is the same serum sample used for qPCR/LAMP and MAT? MAT is usually still negative when Leptospira DNA is detectable in blood, so I am wondering if MAT was not done using a second, convalescent serum sample? This should be made clear.
o line 105, the primer is LipL32-286R (not 268R)
· In the results section, the statement that there was a higher proportion of diabetic patients in severe than in non-severe cases is not supported by the data (15.2% = 5/44 vs 9.1% = 1/11; p=1 Fisher exact test) and is also not shown in the Table 1.
· Line 225: what does "serovarcan" stand for?
Author Response
My major comments are as follow:
1. If my understanding is correct, the design is a comparison of severe leptospirosis cases to non-severe leptospirosis cases. If so, the M&M section must be corrected, since controls must also have a positive test for leptospirosis by any test. In lines 86-87, the statement is that controls were "negative by all tests".
Response Already corrected in line 118 and 120
2. The interpretation of MAT results is largely incorrect and must be completely revised and corrected. First, MAT is a not a serovar-specific technique, but has only some degree of serogroup-specificity. Therefore, results should be presented as serogroups. The text and table headings and legends need only to change serovar to serogroup and Bratislava (serovar) to Australis (corresponding serogroup), because all other names are eponymous serogroups. In the interpretation of MAT results, (Table 2), the total number cannot be higher than the number of patients (=44), because MAT can either
a. point to one putative infecting serogroup (the one giving the highest titre)
b. provide no information on the serogroup, if more than one pathogenic serogroup have the same high titre.
Consequently, the sentence (lines 184-185) stating that one patient was infected with a mixture of
seven serovars is meaningless and must be removed.
Response Already revised and corrected as shown in Table 2
3. The criteria used to define severity should be discussed and compared with the criteria used in similar studies. In another study (PMID 23326614), severe cases were those who died or required vasoactive amines, respiratory support or dialysis. Here, conjonctival suffusion alone or acute hepatic damage (ALAT/ASAT) are considered as markers of severity. Also the findings should be compared to former studies. Here too, PMID 23326614 found that cigarette smoking was a risk (not found here) as was serogroup Icterohaemorrhagiae (not considered here, but could be considered once MAT data is corrected). A major contributor to severity through several studies is the time from onset to antibiotherapy, which is not considered but should be discussed.
Response Already revised in line between 309 and 314, but serogroup Icterohaemorrhagiae related to severity has already stated in line between 272 and 275. A limitation of a variable of antibiotherapy was stated in line between 315 and 317
Minor comments are listed below:
· The introduction could also mention Severe Pulmonary Haemorrhage Syndrome, a very severe form of leptospirosis. (in the 1st paragraph by line 45?).
Response Already added in line 45
· Line 50, the sentence "In return, the incidence rate decreased" is surprising, since the causality was not clearly demonstrated, at least to my knowledge. “In return” should be deleted.
Response Already revised in line 53
· If the design is comparing severe cases to non-severe cases, the phrasing could be improved. e.g. lines 62-63 "... risk factors associated with leptospirosis severity". Also in text and once the Case-control design is presented, the phrasing could use "severe cases" and "non-severe cases" instead of "cases" and "controls"
Response Already revised in line 85 and all phrases in the contents
· The lab methods are not always clear.
o Line 94-95, the blood is collected in a tube of semi-solid culture medium before separation of the serum? Why a culture medium and if so, what culture medium?
Response Already revised in line between 131 and 133. Culture medium is for Leptospira spp. culture . The blood samples were cultured in EMJH semi–solid media
o Is the same serum sample used for qPCR/LAMP and MAT? MAT is usually still negative when Leptospira DNA is detectable in blood, so I am wondering if MAT was not done using a second, convalescent serum sample? This should be made clear.
Response Whole blood for qPCR/LAMP and serum for MAT. We did not provide a protocol of convalescent serum in this study
o line 105, the primer is LipL32-286R (not 268R)
Response Already revised
· In the results section, the statement that there was a higher proportion of diabetic patients in severe than in non-severe cases is not supported by the data (15.2% = 5/44 vs 9.1% = 1/11; p=1 Fisher exact test) and is also not shown in the Table 1.
Response Already corrected
· Line 225: what does "serovarcan" stand for?
Response Already corrected
Round 2
Reviewer 1 Report
I thank the authors for responding to the requests and improving their text. I would suggest some minor improvement of Figure 1:
- Need a scale bar.
- The title could be: “Location of the 12 hospitals-based sites in Thailand, 2017”
- The legend could be: “Hospital location”
- Add the “Province borders” to the legend
- Possibly, the map could show the start of neighboring country borders
Author Response
I would suggest some minor improvement of Figure 1:
- Need a scale bar.
- The title could be: “Location of the 12 hospitals-based sites in Thailand, 2017”
- The legend could be: “Hospital location”
- Add the “Province borders” to the legend
- Possibly, the map could show the start of neighboring country borders
Ans. Already revised as shown in Figure 1
Reviewer 2 Report
It is appreciable that authors have taken into consideration of comments to improve the content and quality of the paper. Systematic and planned rodent control may be appropriate than rodent extermination! Limitations of the study is declared.
Author Response
Systematic and planned rodent control may be appropriate than rodent extermination!
Ans Already revised as shown in line 50
Reviewer 3 Report
The comments raised have been addressed properly.
The description of the methods, notably for Leptospira culture and isolation could be improved. It is very unusual to collect the blood sample directly on a semi-solid culture medium.
In addition, there is no result provided on culture and isolation. In case no Leptospira isolate could be obtained from the blood cultures, I would suggest to either delete the corresponding methods or briefly indicate this result in the results section.
Author Response
In addition, there is no result provided on culture and isolation. In case no Leptospira isolate could be obtained from the blood cultures, I would suggest to either delete the corresponding methods or briefly indicate this result in the results section.
Ans Already added in line 206